# Encapsulation of Luminescent Gold Nanoclusters into Synthetic Vesicles

**DOI:** 10.3390/nano12213875

**Published:** 2022-11-02

**Authors:** Regina M. Chiechio, Solène Ducarre, Célia Marets, Aurélien Dupont, Pascale Even-Hernandez, Xavier Pinson, Stéphanie Dutertre, Franck Artzner, Paolo Musumeci, Célia Ravel, Maria Jose Lo Faro, Valérie Marchi

**Affiliations:** 1Institut des Sciences Chimiques de Rennes, CNRS UMR 6226, Université Rennes 1, F-35000 Rennes, France; 2Dipartimento di Fisica e Astronomia “Ettore Majorana”, Università Di Catania, Via Santa Sofia 64, 95123 Catania, Italy; 3IMM-CNR, Via S. Sofia 64, 95123 Catania, Italy; 4BIOSIT, Inserm, CNRS UMS 3480, Université Rennes1, US_S 018, F-35000 Rennes, France; 5Microscopy Rennes Imaging Centre, SFR Biosit, CNRS UMS 3480—US INSERM 018, Université Rennes 1, F-35000 Rennes, France; 6Institut de Physique, CNRS UMR 6251, Université Rennes 1, F-35000 Rennes, France; 7Service de Biologie de la Reproduction-CECOS, CHU Rennes, F-35000 Rennes, France; 8Irset (Institut de Recherche en Santé, Environnement et Travail), Inserm, EHESP, Université Rennes 1, F-35000 Rennes, France

**Keywords:** gold nanoclusters, vesicles, encapsulation

## Abstract

Gold nanoclusters (Au NCs) are attractive luminescent nanoprobes for biomedical applications. In vivo biosensing and bioimaging requires the delivery of the Au NCs into subcellular compartments. In this view, we explore here the possible encapsulation of ultra-small-sized red and blue emitting Au NCs into liposomes of various sizes and chemical compositions. Different methods were investigated to prepare vesicles containing Au NCs in their lumen. The efficiency of the process was correlated to the structural and morphological aspect of the Au NCs’ encapsulating vesicles thanks to complementary analyses by SAXS, cryo-TEM, and confocal microscopy techniques. Cell-like-sized vesicles (GUVs) encapsulating red or blue Au NCs were successfully obtained by an innovative method using emulsion phase transfer. Furthermore, exosome-like-sized vesicles (LUVs) containing Au NCs were obtained with an encapsulation yield of 40%, as estimated from ICP-MS.

## 1. Introduction

Gold nanoclusters (Au NCs) have appeared as a recent class of non-toxic fluorophores. Their good biocompatibility and their brightness and a large window of photoluminescence lifetime (1 ns–1 μs) make them a promising alternative as fluorescent nanoprobes for biological labeling, bioimaging [1], and in vivo biosensing [2,3]. They are attractive dual nanoprobes that can be detected both by electron microscopy and by luminescence intensity in biological environments. Their ultra-small size (<2 nm) approaches the Fermi wavelength of electrons between metal atoms and nanoparticles. It results in molecule-like properties, including discrete energy levels and the size-dependent fluorescence of the specific electronic structures. Thanks to its high electron density, the presence of the gold element can be confirmed by EDX (Energy-Dispersive X-ray analyses), and the nanoclusters are easily detected by high-resolution electron microscopy imaging. In addition, their ultra-small size facilitates their clearance when they are injected into a living organism. In particular, gold nanoclusters are able to be filtered in the body from the kidneys and urinary excretion rapidly, subsequently reducing their accumulation in the liver or spleen [4,5,6]. Multifunctional nanoconjugates based on Au NCs have been applied in tumor imaging [6].

All the potential applications, including drug delivery, theranostic or biosensing and bioimaging, require transporting the gold nanoclusters into subcellular compartments in vivo. The aim is to deliver them without any alteration of their optical and colloidal properties in view to probe the dynamics of the microenvironment activity or to detect some specific biomolecular targets. Besides the numerous possible nanomaterials used as cargos, liposomes and polymersomes are most commonly used to transport drugs or nanoprobes. Even if they suffer from a lack of control in the release of their contents, the liposomes offer a promising alternative to the polymers to mimic the fluidity of the cellular membrane and their viscoelastic properties [7]. The chemical nature of the lipids used permits them to overcome their endosomal sequestration in the cell, as shown for cationic DOTAP/DOPC fusogenic liposomes [8,9]. They are efficiently used as cargos to deliver drugs into cells [10,11] and extracellular vesicles [12,13]. Recently, efforts have been focused to rationalize the mechanism of the interaction between the lipidic bilayers and nanoparticles (NPs), such as magnetic maghemite nanoparticles, quantum dots [14], or hydrophobic gold nanoparticles [15,16,17,18], depending on the fluidity of the lipidic membrane [19] and the NP size [20]. From this point of view, the size of the gold nanoclusters is expected to be advantageous as they should not induce leakage or disruption of the membrane [21]. In view to deliver the luminescent Au NCs into the subcellular compartments, it is then required to investigate how to associate the gold nanoclusters with the lipidic bilayers of the extracellular vesicles [13,21] or cells. We previously demonstrated that the gold nanoclusters with a controlled surface chemistry and charge can strongly electrostatically interact with the headgroups of the phospholipidic bilayer without disrupting the vesicle structure in the presence of an excess of vesicles [21]. It is of great importance to investigate how to incorporate them in the vesicle lumen to vehicle them inside the cells by adjusting their interaction, thanks to the easy Au surface chemistry. Furthermore, the NCs’ encapsulation into the vesicles is also expected to avoid interactions with blood proteins or other fluid components and prolong the circulation of the cargos into the organism by adjusting the chemical nature of the lipidic components of the membrane, as demonstrated, for example, with stealthy PEGylated liposomes [22].

Here, we explore the possibility of encapsulating hydrophilic gold nanoclusters into synthetic liposomes by the optimization of their surface chemistry and the preparation method of the liposomes. We evaluated the encapsulation capacity of the liposomes depending on their size and the chemical coating. Several preparation methods were investigated to determine how gold nanoclusters are associated with the lumen of vesicles. We have described successful methods to prepare cell-like-sized vesicles (GUVs or MLVs) and exosome-like-sized vesicles (LUVs) with the quantification of the encapsulation of the Au NCs’ yield. The efficiency of the process was correlated to the structural and morphological aspects of the Au NCs encapsulating vesicles.

## 2. Materials and Methods

C_3_E_6_D Red NCs Synthesis [21]: All glassware used for these syntheses was cleaned in a bath of freshly prepared aqua regia (HCl:HNO_3_, 3:1 in volume) and rinsed in water 10 times before using them. An aqueous solution of glutathione (50 mM, 1.2 mL), denoted GSH, in ultrapure water (16.8 mL) was heated at reflux at 120 °C under stirring. Then, hydrogen tetrachlorate trihydrate HAuCl_4_, 3 H_2_O (20 mM, 2 mL) was rapidly added. The reaction was stopped by cooling the mixture after 3 h. The concentration of the aqueous suspension of orange-emitting GSH Au NCs was found to be 2.6 μM by ICS-MS measurement, as described below. Then, a solution of GSH Au NCs suspension (2.6 μM, 1 mL, 1 mol eq) was incubated in the dark in the presence of an aqueous solution of C_3_E_6_D (20 mM, 0.155 mL, 5 mol eq) at room temperature overnight. The C_3_E_6_D red NCs’ solution obtained could be stored at 4 °C for months without significant change in their optical properties.

C_3_E_6_D Blue NCs Synthesis: All glassware used for these syntheses was cleaned in a bath of aqua regia (HCl:HNO_3_, 3:1 in volume) and rinsed in water 10 times before using them. An aqueous solution of C_3_E_6_D (20 mM, 0.585 mL) in ultrapure water (2.7 mL) was heated at reflux at 120 °C under stirring. Then, hydrogen tetrachlorate trihydrate HAuCl_4_, 3 H_2_O (20 mM, 0.3 mL) was rapidly added. The reaction was stopped by cooling the mixture after 4 h.
C_3_E_6_D Blue or Red NCs were purified from the C_3_E_6_D excess by ultrafiltration (Amicon 3KDa, Merck, Darmstadt, Germany) at 17,000 g for 10 min. What remained inside the filter and did not pass the 3 KDa membrane would be the purified C_3_E_6_D Blue or Red NCs.

Giant unilamellar vesicles (GUVs) preparation [23]: A lipidic solution was prepared by dissolving a pure DOPC or (DOPC:DOTAP) (95:5) molar lipidic mixture in chloroform (final lipid concentration, 13.3 mM). A volume of 20 μL  of this lipidic solution $\left(0.27\;\mu \mathrm{mol}\right)$ was mixed with 1.8 mL of paraffin oil and heated at 80 °C for 30 min. The traces of chloroform were removed by evaporation under vacuum to give a final lipid solution (total lipid concentration, $10^{-4}\;M$). To prepare the GUV, a volume of 50 μL  of an aqueous sucrose solution (500 mM) was added to 400 μL of the lipid solution (total lipid concentration, $10^{-4}\;M$) and vortexed for 40 s to form a water-in-oil (w/o) emulsion. Then, this emulsion was gently added on top of a sucrose solution of high viscosity (200 μL, 500 mM in a second Eppendorf) without mixing. After 10 min, the solution was centrifuged for 15 min at 18,890× *g*. The bottom solution was transferred into another Eppendorf tube, redispersed in 300 μL  of a glucose solution (500 mM), and centrifuged again at 18,890× *g* for 5 min. After centrifugation, the bottom solution was taken up, and the GUVs were ready to be stored at 4 °C.

NCs encapsulated into Giant Unilamellar Vesicles (GUVs): To encapsulate the NCs in the GUVs, 25 µL of a sucrose solution (1 M) and 25 µL of an NCs solution (particle concentration, 2 μM) were added to 400 μL of a lipid solution in chloroform (total lipid concentration, $10^{-4}\;M$) and then vortexed for 40 s to form a water-in-oil (w/o) emulsion. This emulsion was gently added on top of a sucrose solution of high viscosity (200 μL, 500 mM, in a second Eppendorf tube) without mixing. After overnight incubation, the GUVs were formed in the bottom and were ready to be collected.

NCs encapsulated into Multilamellar Vesicles (MLVs) and Large Unilamellar Vesicles (LUVs): A lipid solution was prepared by dissolving a DOPC or a DOPC: DOTAP (95:5) lipidic mixture in chloroform ($10\;\mathrm{mg}\cdot {\mathrm{mL}}^{-1}$, total lipid concentration, 13.3 mM). A lipidic film was formed from the evaporation of 1 mL of this solution using a rotary evaporator at 50 °C and 70 mbar. After the introduction of a mixture of C_3_E_6_D NCs (500 μL, NC concentration around 2 μM) and glucose (500 μL, 500 mM) into a flask, the hydrated phospholipidic film was successively immersed in liquid nitrogen for 30 s and then in a water bath at 50 °C for 30 s. This cycle was repeated 5 times to form the NC-encapsulated MLV suspension. The NC-encapsulated MLVs were then extruded 10 times at a maximal pressure of 15 bars using a lipex Biomembranes Inc apparatus (Vancouver, BC, Canada), according to the literature [24]. At the end of this procedure, the LUVs suspension was purified by ultrafiltration (Amicon 30 kDa).

Small-Angle X-ray Scattering (SAXS): X-ray scattering results were collected with a Pilatus 300K (Dectris AG, 5405 Baden-Daettwil, Switzerland) and mounted on a micro source X-ray generator GeniX 3D (Xenocs, Grenoble, France) operating at 30 W. The monochromatic CuK_α_ radiation was *λ* = 1.541 Å. The sample to the detector distance (273 mm) was calibrated using silver behenate. The results were recorded in a reciprocal space where q = (4πsinθ)/*λ* and where θ is the diffraction angle, in a range of repetitive distances from 600 to 3.6 Å. The results were collected by a homemade program and analyzed by Igor Pro 7.0 software (Wave metrics, Portland, OR, USA). The acquisition time was 1 h. Samples were loaded in thin Lindman glass capillaries (with a diameter of 1 ± 0.1 mm and a thickness of 10 µm; GLAS, Muller, Berlin, Germany) sealed with paraffin. The lipid–NCs hybrid complexes were prepared by a mixture of a micromolar concentration NCs solution (10 µL, 1.70 µM NCs concentration) and millimolar concentration SUV suspension (10 µL, 16 mM total phospholipid concentration) in the glass capillaries. All samples exhibited powder diffraction rings, and the scattering intensities as a function of the radial wave vector were determined by circular integration.

Inductively Coupled Plasma Mass Spectroscopy (ICP-MS): The concentrations of gold (Au) and Phosphorus (P) elements in the vesicles’ suspension and in the AuNC suspension were determined by using the Inductively Coupled Plasma Mass Spectroscopy (ICP-MS) technique. The calibration curves for Au and P from 0.2 to 2 ppm were established from standard commercial solutions. The pure red and blue AuNC were filtered from the free ligands and residual gold salt excess by ultrafiltration (Amicon 3 KDa) at 17,000 g for 10 min. The NC suspensions were purified on a size-exclusion gel chromatography column (NAP-5 Cytiva). The dissolution of the Au NCs was then obtained by incubation in the presence of aqua regia (1 HNO_3_:3 HCl) for 12 h. Different volumes of the starting solution (65 μL, 125 μL, 250 μL) were added to 500 μL aqua regia and left at 4 °C overnight. In each case, the Au concentration was compared with a calibration curve established from a standard commercial Au solution. The AuNC concentration was then deduced from the Au concentration given the number of Au atoms per AuNC estimated from the emission wavelength, and the diameter was obtained by electron microscopy.

The liposome disruption and the dissolution of the Au NCs contained inside were realized by incubation in the presence of aqua regia for 12 h. For each sample of AuNC and the LUV or MLV mixture, different volumes (65 μL, 125 μL, 250 μL) were added to 500 μL aqua regia and heated for 1 h at 50 °C to better dissolve the lipids and at the left at 4 °C overnight. To obtain a precise measurement, three solutions at different concentrations of NCs encapsulated in the LUVs were prepared to be analyzed. The next day, H_2_O was added to bring the volume of each sample to a total of 25 mL, and the measurements were carried out and compared to the corresponding calibration curves.

Spectrofluorimetry: Photoluminescence measurements were performed on a Jasco FP-8300 spectrofluorometer. The measurements were performed at room temperature on liquid samples. The wavelength resolution of both the excitation and the emission slits was set to 5 nm, the response times were 0.5 s and the scan speed was 500 nm/min. The absolute quantum yields were measured with a C9920–03 Hamamatsu system.

UV-visible spectroscopy: The UV-visible absorbance measurements were performed using a ThermoFisher Scientific NanoDrop by placing 2 μL of a non-diluted sample over a pedestal.

Dynamic Light Scattering (DLS): The measurements of the mean hydrodynamic diameters were performed at an angle of 90° using a Nanosizer ZEN3600 (Malvern Instruments, Worcester, UK) and collected at 25 °C.

Light Optical Microscopy: The fluorescence optical microscopy observations were performed either under direct bright light or epifluorescence on an inverted microscope IX71 (Olympus, Tokyo, Japan) equipped with a 20× objective (NA = 0.45) (Olympus, Japan). Excitation was provided at 365 nm with a high vacuum mercury lamp (200 W). Images were acquired by a Photometrics CoolSNAP HQ2 camera equipped with a soft imaging system (Olympus, Tokyo, Japan).

Confocal Microscopy: Fluorescence confocal images were acquired using a LEICA SP8 confocal microscope equipped with a 63× oil immersion objective (NA = 1.40). Excitation light was provided with a 405 nm laser diode.

Transmission Electron Microscopy (TEM): Transmission electron microscopy analyses were carried out with a JEOL 2100 transmission electron microscope operated at 200 KV supplied with an UltraScan 1000XP CCD camera. For the sample preparation, 300 mesh carbon-coated nickel grids were placed for 1 min on top of a 40 µL sample droplet and dried up with paper. Particle sizes were determined from TEM micrographs using Fiji Software.

Cryo-Transmission Electron Microscopy (Cryo-TEM): Vitrification of the vesicles was performed using an automatic plunge freezer (EM GP, Leica) under controlled humidity and temperature [25]. The samples were deposited to glow-discharged electron microscope grids followed by blotting and vitrification by rapid freezing into liquid ethane. Grids were transferred to a single-axis cryo-holder (model 626, Gatan) and were observed using a 200 kV electron microscope (Tecnai G^2^ T20 Sphera, FEI) equipped with a 4k × 4k CCD camera (XF416, TVIPS). Micrographs were acquired under low electron doses using the camera in binning mode 1 and at a nominal magnification of 25,000×.

## 3. Results and Discussion

### 3.1. Synthesis and Characterization of the Red and Blue Gold Nanoclusters (Red and Blue NCs)

In view of exploring the role of the surface chemistry, two types of NCs, C_3_E_6_D Red NCs and C_3_E_6_D Blue NCs, were synthesized. The C_3_E_6_D ligand permits it to have strong adhesion to the Au surface through the three thiol groups of cysteine but also allows it to increase stability against aggregation thanks to the steric stabilization of the polyethyleneglycol (PEG) spacer and the electrostatic repulsive stabilization of the two negatively charged carboxyl groups. Furthermore, it also allows the turning of the NCs into a multivalent platform by means of the two carboxylic groups, which can react with antibodies or other chemical recognition groups such as the peptides of the biological interest on the surface of the NCs for theranostic [26,27]. To create the Red NCs, two steps were necessary: At first, the NCs were synthesized by the reduction of the gold salts with glutathione (GSH) [21]. Then, the second step of ligand exchange was performed to functionalize them with the peptide ligand, C_3_E_6_D, as shown in Figure 1a,b. For the Blue NCs, on the other hand, a one-step synthesis was used, which allowed the simultaneous nucleation and functionalization of the NCs with the C_3_E_6_D. The morphological and optical characterizations are presented in Figure 1c–e, respectively. In the case of the Red NCs, the diameter evaluated by TEM was about 2.4 ± 0.5 nm. The optical excitation and luminescence intensity spectra of the red NCs exhibited an excitation peak at 420 nm and an emission peak in the red region with a maximum intensity at 620 nm. Instead, in the case of the Blue NCs, there were smaller particles with an average diameter of 1.7 ± 0.5 nm and with an excitation peak at 410 nm and an emission peak in the blue region at 490 nm. The absolute luminescence yields were found to be 4% for the Blue NCs and 2% for the Red NCs, which is in agreement with the literature [1,2].

### 3.2. Red and Blue NCs’ Encapsulation into Giant Unilamellar Vesicles (GUVs)

Once the Au nanoclusters were well-characterized, an innovative encapsulation technique was developed (Figure 2a) using an emulsion phase transfer, as reported by Weitz et al. [28] and adapted from a recent protocol described in the literature for the formation of GUVs [23]. An initial w/o (water-in-oil) emulsion was prepared by vigorously stirring a small quantity of the aqueous solution containing the C_3_E_6_D NCs to a solution of the lipidic mixture in chloroform, effectively forming water droplets coated with the lipids. Then, the emulsion was poured on top of an oil-in-water biphasic system where the lipids dissolved in the chloroform should have assembled into a monolayer at the interface. Due to their density difference, the monolayer-coated water droplets from the emulsion sunk to the aqueous phase through the lipidic saturated interface, generating (w/o/w) double emulsions. Water droplet migration was facilitated by using 500 mM of sucrose in the inner water droplet and iso-osmotic aqueous 500 mM glucose in the biphasic system and by doing further centrifugation, which can remove the trapped organic solvent. Then, this emulsion was gently poured on top of the bottom aqueous solution. Giant vesicles with a high contrast were observed by optical phase contrast microscopy, as expected from the difference in the refractive index between the glucose and sucrose (see Figure 2b,d). The vesicles appeared to be fluorescent, in agreement with the encapsulation of the Au nanoclusters in their lumen (see Figure 2c,e). This method clearly resulted in the formation of GUVs composed of a double lipid layer, encapsulating the NCs.

Several experiments were performed to better understand this result. Initially, for simplicity, only the Red NCs were used to optimize the conditions and to study the various parameters, such as the surface charge of the vesicles and NCs, the surface ligand used, and the chemical nature and the concentrations of lipids and NCs involved. In particular, negatively charged NCs (C_3_E_6_D Red NCs) were encapsulated inside neutral vesicles or positively charged vesicles, leading to, in both cases, an efficient encapsulation, whatever the vesicle surface charge (Appendix A). On the contrary, the type of NCs’ surface ligand affected the encapsulation efficiency. Indeed, by using GSH NCs instead of C_3_E_6_D NCs, GUVs encapsulating NCs were formed, but the encapsulation efficiency was lower, as shown qualitatively from the high level of background fluorescence in Figure 2e. This result could be attributed to the stronger interaction of the GSH NCs with the lipid membrane, which tends to aggregate the vesicles. This interaction could also induce bilayer destabilization and leakage, resulting in the release of the NCs into the external medium.

To better visualize the luminescence of the NCs, the GUVs containing NCs formed were observed by fluorescence confocal microscopy (Figure 3). In this experiment, we used C_3_E_6_D Blue (Figure 3a–c) or Red (Figure 3d–f) NCs. In both cases, luminescent disks, which coincided with the vesicles in the phase contrast, were observed. Images were recorded and are shown in the Appendix A for a more comprehensive view (Appendix A) and also a video is available in z-stack (Appendix A). One can notice different levels of luminescence intensity from one to another of the GUVs, which could be attributed to the multivesicular structure within a given GUV. These observations clearly indicate that the NCs have been encapsulated within the vesicles without disrupting them or altering their integrity.

### 3.3. Red and Blue NCs’ Encapsulation into Multilamellar (MLVs) and Unilamellar (LUVs) Vesicles

To obtain a more quantitative evaluation of the encapsulation efficiency, DOPC multilamellar vesicles (MLVs) in the presence or not of the NCs in an aqueous solution were produced according to well-known processes. Briefly before the formation of the vesicles, the lipids were dissolved in chloroform, which was evaporated entirely to form a thin layer in the vial. Then, the addition of the aqueous medium (containing the NCs or not) caused swelling of the film and ultimately formed vesicles once the bilayer left the solid surface of the vial. The swelling and budding stage was facilitated by mechanical stirring and five freezing and thawing cycles. As expected from this method, the film rehydration generated vesicles with broad size distributions, as shown in the DLS measurements (Figure 4a). [29,30]

The cryoTEM images have confirmed the multilamellar morphology of the pure MLVs (Figure 5a,b) and also in the presence of the NCs (Figure 5c,d). The onion-like structure with multiple internal aqueous compartments induces a strong contrast that complicates the visualization of the NCs. However, it is possible to distinguish, on the background, the presence of small black dots that can be attributed to small aggregates of the NCs (Figure 5c, Appendix A), which are not visible in the control sample of the pure MLV without the NCs (Figure 5d). The presence of the NCs may induce the deformation of the MLV membranes and clustering at the level of the lipid bilayers and is, consequently, visible only when they form small aggregates (see also Appendix A). Then, sequential extrusion is used to achieve better distribution control [24,30]. This technique consists of repeatedly passing a crude heterogeneous mixture of vesicles (mostly MLVs) through a polycarbonate membrane with small pores of a 100 nm diameter to obtain homogeneously distributed vesicles of a diameter matching the pore size, as shown by DLS in Figure 4b. As expected from the extrusion process with the 100 nm pore size, [24] most of all the vesicles are unilamellar, as shown by cryoTEM (Figure 6a). As previously observed, the extrusion process induces their deformation, and non-spherical unilamellar vesicles are also present (Appendix A) [29]. The NCs are not easy to image over the LUVs’ membrane. Nevertheless, it is possible to distinguish dark points between the bilayers that could be attributed to the AuNCs. The dark small aggregates are localized where the vesicles are in contact from one to the other, forming several stacked lipid bilayers (Figure 6b–d).

To investigate more precisely the effect of the NCs on the bilayers’ organization, small angle X-ray scattering measurements (SAXS) were recorded for the MLVs and LUVs. The MLVs’ spectra (Figure 7a) exhibit two well-defined peaks at $0.097\;{Å}^{-1}\;$ and $0.192\;{Å}^{-1}\;$, corresponding to the first and second order of a multilamellar stacking, the first of which is attributed to the lamellar repeat distance i.e., the sum of the DOPC bilayer thickness and of the water gap. In the presence of the NCs, the intensities of the two peaks are lowered. As the NCs are positioned in the aqueous compartments confined in between the adjacent bilayers, they induce disorder in the multilamellar stacking. This organization modification in the presence of the NCs explains why fewer MLVs were observed by the cryoTEM, as well as fewer lamellar aggregates observed by the UV-Vis spectra (Appendix A) and DLS techniques (Figure 4). In the case of the LUVs (Figure 7b), the lamellar orders vanish into the diffuse scattering of individual membranes [21] with a broad peak at $0.13\;{Å}^{-1}$. This 49 Å corresponds to the interdistance between the high electron density parts of the membrane, which are the phosphate headgroup in the case of the DOPC. As the NCs do not interact with the membranes, they can be eliminated by size exclusion chromatography (Figure 7b).

To quantify the efficiency of the NCs’ encapsulation, the luminescence spectra for the MLVs and LUVs before and after the elimination of the NCs were first recorded (Figure 8). The presence of the typical emission peak at 630 nm confirms that the NCs’ presence and the intensity decrease that was observed following the purification process gave a first estimation of the encapsulation yield. However, the emission intensity was not only related to the amount of NCs present in the vesicle lumen. In particular, the NCs’ emissions could be affected by the microenvironment. In view of better quantifying the encapsulation yield of the NCs, the concentrations of the gold and phosphorus elements were determined by inductively coupled plasma mass spectroscopy (ICP-MS) in the vesicular suspensions before and after extrusion (MLV or LUVs) and before and after the ultrafiltration process to eliminate the free NCs in excess (Table 1). The MLVs and LUVs after purification had almost the same Au concentration (respectively $31.88\;\mathrm{mg}\cdot L^{-1}\;$ and $30\;\mathrm{mg}\cdot L^{-1}$), indicating that in both the LUVs and the MLVs, most of the NCs were located in the vesicle lumen. Indeed, the extrusion process resulted in the peeling of the onion-like structure of the MLVs until they had a unilamellar vesicle, defined as an aqueous compartment closed by one single bilayer. Regarding the phosphorus concentration, only 18% of the initial amount of phosphorus used to prepare the MLVs was found in the suspension of the LUVs encapsulating the NCs. This observation was in agreement with the loss of the bilayers resulting from the extrusion process [30]. Moreover, the Au encapsulation yield was found to be 40% when considering the initial Au concentration ($74\;\mathrm{mg}\cdot L^{-1}$) compared to that of after the extrusion and purification ($30\;\mathrm{mg}\cdot L^{-1}$) of the MLVs in the presence of the NCs. This value is higher than the typical 10% volume ratio of the trapped volume expected from extruded LUVs with a pore size of 100 nm and a lipid concentration of 10 $\mathrm{mg}\cdot {\mathrm{mL}}^{-1}$, according to the literature [24]. Hence, the Au encapsulation yield of 40%, here, appears higher than that of the trapping capacity of the liposomes. This latter result suggests that the NCs were not only trapped in the internal volume of the liposomes but could also be embedded in the bilayer membrane. This is also in agreement with the cryoTEM images, where small aggregates of the Au NCs appeared to be adsorbed on the vesicle membrane, thus greatly increasing the Au trapping efficiency of the liposomes.

## 4. Conclusions

In conclusion, the synthesis of red Au NCs was optimized to obtain an aqueous stable GSH NCs’ suspension in only 3 h with NCs’ dimension of about 2 nm and with a fluorescence excitation peak at 430 nm and an emission peak at 620 nm. Then, a ligand exchange performed with C_3_E_6_D here allowed the grafting of the NCs through thiol groups, which could be extended to other peptide derivatives to functionalize the NCs. The high colloidal stability of the C_3_E_6_D NCs and their ultra-small size made it possible to encapsulate the C_3_E_6_D NCs into the GUVs or LUVs with high efficiency by preserving the bilayer integrity and the vesicle morphology. Thanks to a detailed structural and chemical analysis, it was possible to demonstrate the efficiency of the innovative encapsulation methods presented above. The encapsulation efficiency was quantified by ICP-MS and found to be 40% for the Au NCs in the LUVs prepared by extrusion. This result was obtained thanks to an optimization of the different parameters, such as the choice of the lipids and the ligands grafted to the NCs’ surface. The use of a pegylated peptide-noted C_3_E_6_D by increasing the stability of the NCs appeared to hinder the membrane rupture and the consequent leakage of the NCs into the external liquid. In the next future work, the chemical nature of the vesicle bilayer will be optimized in order to introduce specific recognition groups that can vehicle the NCs to target organism parts with a high concentration. Therefore, the ultra-small and stable gold nanoclusters could serve not only as biomarkers of lipidic membranes. Their encapsulation in the vesicle lumen used as cargos could bring Au NCs into the cells or EVs for in situ biosensing [31] or drug delivery [32].

## Figures and Tables

**Figure 1 nanomaterials-12-03875-f001:**
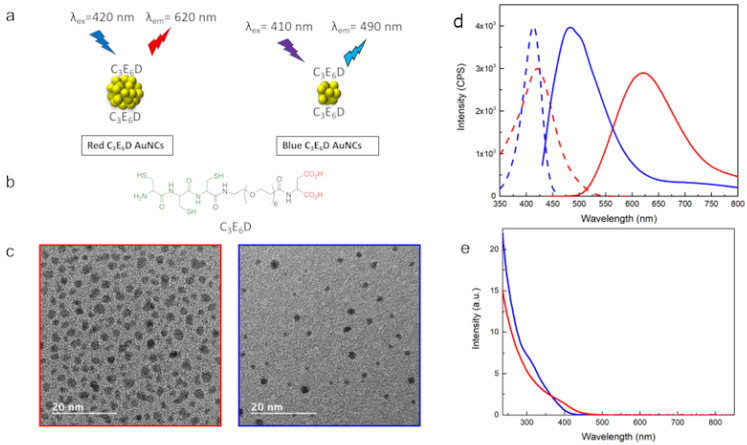
(**a**) Schematic view of Red and Blue NCs. (**b**) Chemical structure of NCs’ ligand C_3_E_6_D. (**c**) TEM micrographs, (**d**) luminescence excitation (dotted lines), and emission spectra (solid lines), and (**e**) absorbance spectra of C_3_E_6_D Blue and Red NCs (blue and red line, respectively).

**Figure 2 nanomaterials-12-03875-f002:**
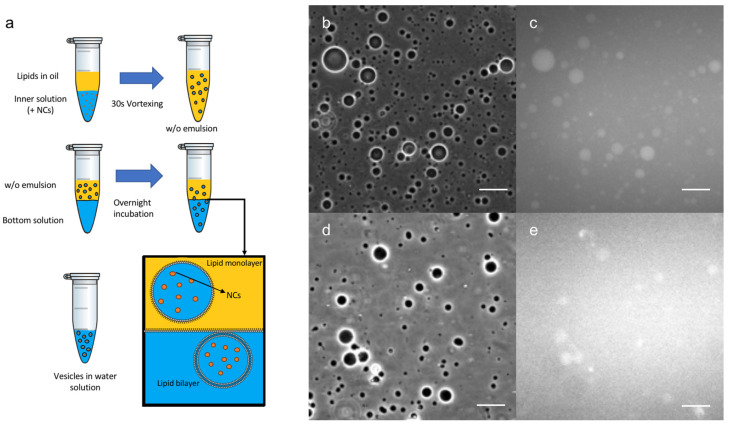
(**a**) Schematic view of Au NCs’ encapsulation method into GUVs. Optical microscope images in phase contrast (**b**,**d**) or luminescence (**c**,**e**) of: (**b**,**c**) C_3_E_6_D and (**d**,**e**) GSH Red NCs’ encapsulated into DOPC GUVs. Scale bar, 50 µM.

**Figure 3 nanomaterials-12-03875-f003:**
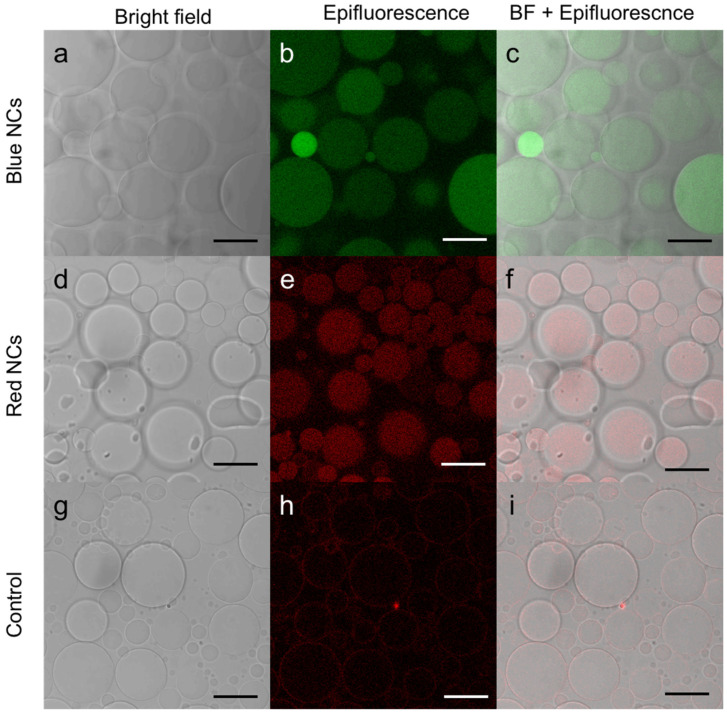
Confocal microscopy image of (**a**–**c**) C_3_E_6_D Blue Au NCs, (**d**–**f**) C_3_E_6_D Red Au NCs’ encapsulated in DOPC GUVs, and (**g**–**i**) DOPC GUVs without Au NCs as a control obtained by (**a**,**d**,**g**) bright field, (**b**,**e**,**h**) luminescence, and (**c**,**f**,**i**) merged images. Scale bar, 20 µM.

**Figure 4 nanomaterials-12-03875-f004:**
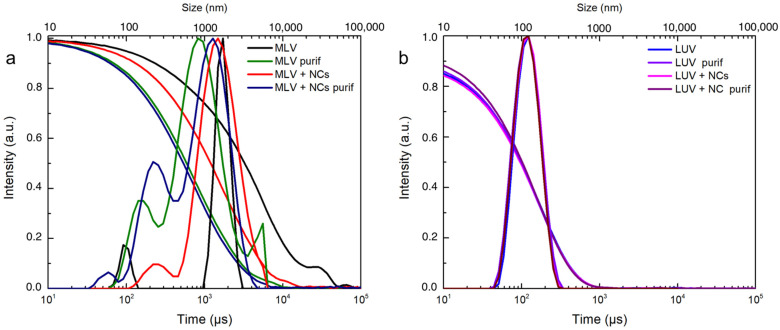
DLS intensity and correlogram of (**a**) MLVs and (**b**) LUVs with or without C_3_E_6_D Red NCs and before and after purification by elimination of free AuNCs.

**Figure 5 nanomaterials-12-03875-f005:**
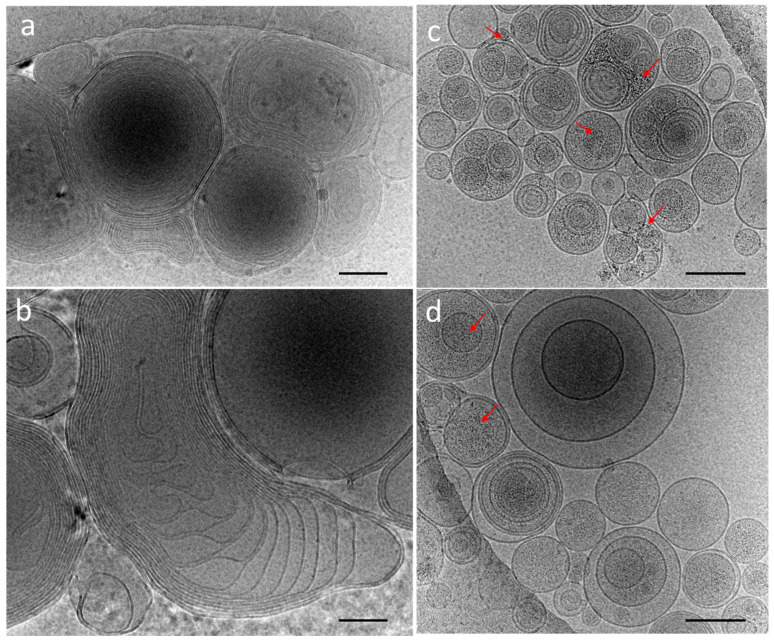
CryoTEM micrographs of DOPC MLVs’ (**a**,**b**) control without NCs and (**c**,**d**) in the presence of C_3_E_6_D Red NCs. The red narrows point out the presence of AuNCs. Scale bar represents 400 nm.

**Figure 6 nanomaterials-12-03875-f006:**
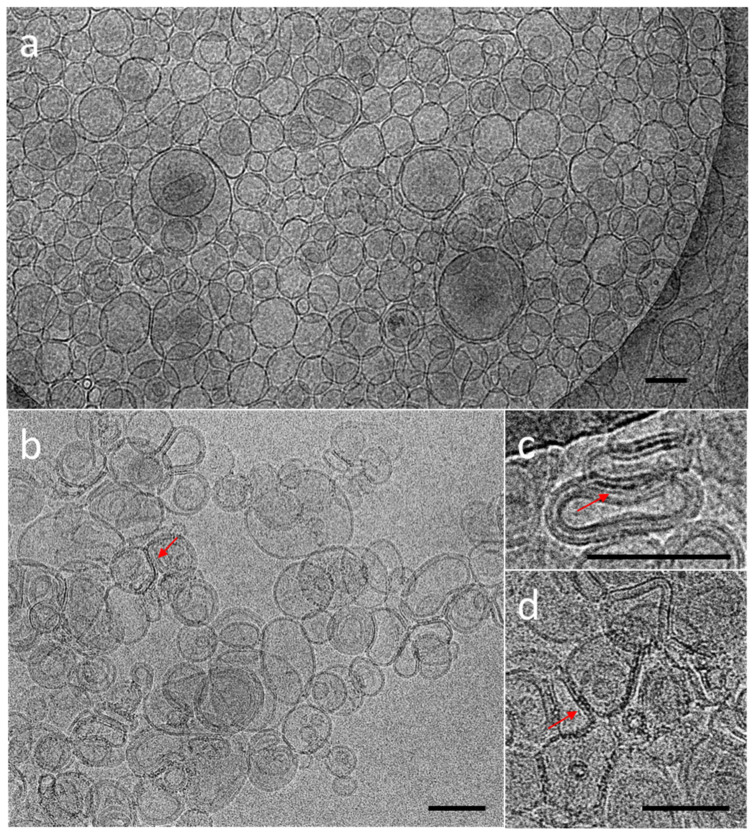
CryoTEM micrographs of DOPC LUVs (**a**) without NCs, (**b**–**d**) in the presence of C_3_E_6_D Red NCs. The red narrows point out the presence of AuNCs. Scale bar represents 100 nm.

**Figure 7 nanomaterials-12-03875-f007:**
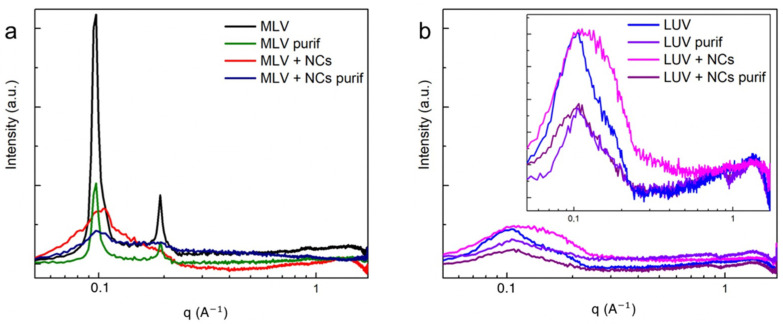
SAXS intensity spectra of (**a**) MLVs and (**b**) LUVs with and without C_3_E_6_D Red NCs, before and after purification.

**Figure 8 nanomaterials-12-03875-f008:**
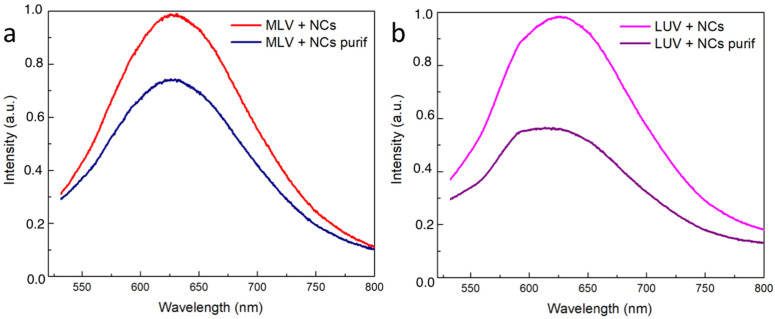
Normalized emission intensity spectra of the vesicles encapsulating the NCs: (**a**) MLVs and (**b**) LUVs before and after elimination of free NCs.

**Table 1 nanomaterials-12-03875-t001:** Gold concentration $\left[Au\right]\;\left(\mathrm{mg}\cdot L^{-1}\right)$ in the suspension of vesicles encapsulating C_3_E_6_D Red NCs before and after purification determined by ICP mass spectroscopy.

MLV+NCs	MLV+NCs Purif	LUV+NCs	LUV+NCs Purif
74∓10	32∓2	54∓1	30∓ 2

## Data Availability

Data available on request from authors.

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
