# Peer review of "Encapsulation of Luminescent Gold Nanoclusters into Synthetic Vesicles"

_nanomaterials, 2022, doi:10.3390/nano12213875_

Round 1

Reviewer 1 Report

Dear Editor,

This is an interesting article dealing with a evaluation of the  possibility to encapsulate hydrophilic gold nanoclusters into synthetic liposomes for preparation of appropriate nanostructures with potential applicability for bioimaging and sensing. The topic of the article fits the scope of the journal and could be of interest for a wide variety of scientists working on preparation of nanostructures for bioapplications. 

Thus, I recommend publication of the article in Nanomaterials but after taking into consideration the following minor issues:

1. Abstract section: authors mention that "Gold nanoclusters (Au NCs) are attractive dual luminescent nanoprobes". However, in the manuscript there is not any evidence of such dual-luminescence behavior. Please revise.

2. The aim of the article is to "study the possibility to encapsulate hydrophilic gold nanoclusters into synthetic liposomes". I agree that this is a very interesting topic and the excellent results in terms of encapsulation efficiency (40%) make this work highly valuable. However, it is not clear the significant advances/differences of the encapsulation process and in the obtained results as compared with some other previous references that have been focused on evaluations of the interaction between lipidic bilayers and nanoparticles such as gold nanoparticles. A critical comparison with such systems will be very welcome.

3. Materials and Methods section. C3E6D Red NCs Synthesis: Authors mention that "The concentration of the aqueous suspension of Au NCs (in µM) was determined by ICP-MS". In this point two important issues must be clarified: first, did authors performed any purification step before ICP-MS Au measurement? How are you sure that all Au is in form of Au NCs? On the other hand, how can you obtain the AuNCs micromolar concentration from the Au measurement by ICP-MS? Please clarify. 

4. How did the authors compute the NCs absolute luminescence quantum yield? Is 4% a value competitive with other Au NCs previously described?

5. Table 1: it would be valuable if the values presented here are accompanied of their statistical deviation. 

End of report

Author Response

We are thankful to the reviewers for all their fruitful comments and suggestions. All the mentioned points have been taken into account as follows.

Review 1

This is an interesting article dealing with a evaluation of the  possibility to encapsulate hydrophilic gold nanoclusters into synthetic liposomes for preparation of appropriate nanostructures with potential applicability for bioimaging and sensing. The topic of the article fits the scope of the journal and could be of interest for a wide variety of scientists working on preparation of nanostructures for bioapplications. 

Thus, I recommend publication of the article in Nanomaterials but after taking into consideration the following minor issues:

  1. Abstract section: authors mention that "Gold nanoclusters (Au NCs) are attractive dual luminescent nanoprobes". However, in the manuscript there is not any evidence of such dual-luminescence behavior. Please revise.

We suppress « dual » from the abstract. We only mention that gold nanoclusters are attractive dual nanoprobes in the introduction because they can be detected by electron microscopy and by fluorescence optical microscopy. In the present paper, we visualize the gold nanoclusters encapsulated in the vesicles by electron microscopy and also by optical microscopy.

  1. The aim of the articleis to "study the possibility to encapsulate hydrophilic gold nanoclusters into synthetic liposomes". I agree that this is a very interesting topic and the excellent results in terms of encapsulation efficiency (40%) make this work highly valuable. However, it is not clear the significant advances/differences of the encapsulation process and in the obtained results as compared with some other previous references that have been focused on evaluations of the interaction between lipidic bilayers and nanoparticles such as gold nanoparticles. A critical comparison with such systems will be very welcome.

We agreed with the reviewer. Nevertheless to the best of our knowledge, there is no published work about the quantitative chemical evaluation of gold nanoclusters amount encapsulated into liposomes. In addition it is rather difficult to compare to gold nanoparticles because their interaction is rather different due to their larger size; they induce deformation, invagination and even destabilization of the lipidic membranes making hazardous any comparison.

  1. Materials and Methods section. C3E6D Red NCs Synthesis: Authors mention that "The concentration of the aqueous suspension of Au NCs (in µM) was determined by ICP-MS". In this point two important issues must be clarified: first, did authors performed any purification step before ICP-MS Au measurement? How are you sure that all Au is in form of Au NCs? On the other hand, how can you obtain the AuNCs micromolar concentration from the Au measurement by ICP-MS? Please clarify. 

Before the ICP-MS titration of Au, the samples were filtered and concentrated on Amicon membrane with a 1kDa cutoff. Then the sample was purified on a size-exclusion gel chromatography column to eliminate any trace of gold precursor, Au(I) complex or ligand in excess.

The Au amount of Au NC suspension was estimated by dissolution of three different given volumes of AuNC with aqua regia. In each case the Au concentration was compared with a calibration curve established from a standard commercial Au solution. The AuNC concentration was then deduced from the Au concentration given the number of Au atoms per AuNC estimated from the emission wavelength and the diameter obtained by electron microscopy.

  1. How did the authors compute the NCs absolute luminescence quantum yield? Is 4% a value competitive with other Au NCs previously described?

The absolute quantum yields were measured with a C9920–03 Hamamatsu system. The 4% QY value of the C3E6D red NCs is slightly higher thant the one of GSH red NCs, i.e. 2% but remains in the same range.

  1. Table 1: it would be valuable if the values presented here are accompanied of their statistical deviation. 

Error bar were added for each value (see the manuscript).

Reviewer 2 Report

The paper discusses the possible encapsulation of ultra-small sized red and blue 18 emitting AuNCs into liposomes of various size and chemical composition, including different methods investigated to prepare vesicles containing AuNCs in their lumen. A correlation between efficiency of the process and the structural and morphological aspect of the Au NCs encapsulating vesicles was done. Various analyses by SAXS, cryo-TEM and confocal microscopy techniques were used. Cell-like sized vesicles (GUVs) encapsulating red or blue Au NCs by emulsion phase transfer and  exosome-like sized vesicles (LUVs) containing uNCs were obtained

I suggest the paper acceptance with a few minor corrections.

Line 85: Please correct: "Firstly, A freshly prepared aqueous solution" 

Line 97 starts with: Firstly, a freshly prepared aqueous solution of ...I would suggest reformulating one of the two sentences.

Reference 3 is the same as reference 1

Author Response

We are thankful to the reviewers for all their fruitful comments and suggestions. All the mentioned points have been taken into account as follows.

Review 2

The paper discusses the possible encapsulation of ultra-small sized red and blue 18 emitting AuNCs into liposomes of various size and chemical composition, including different methods investigated to prepare vesicles containing AuNCs in their lumen. A correlation between efficiency of the process and the structural and morphological aspect of the Au NCs encapsulating vesicles was done. Various analyses by SAXS, cryo-TEM and confocal microscopy techniques were used. Cell-like sized vesicles (GUVs) encapsulating red or blue Au NCs by emulsion phase transfer and  exosome-like sized vesicles (LUVs) containing uNCs were obtained

I suggest the paper acceptance with a few minor corrections.

Line 85: Please correct: "Firstly, A freshly prepared aqueous solution" 

Line 97 starts with: Firstly, a freshly prepared aqueous solution of ...I would suggest reformulating one of the two sentences.

The sentences were rewritten.

Reference 3 is the same as reference 1

The references were corrected.

We also insert the video S3 and the figure S7 that were missing.

The  figure 5 was completed with additional cryoTEM images (see Figure 5c and 5d; Figure S5).

Prof. Celia Ravel was forgotten as author and therefore added.